# Quality Characteristics of Karst Plateau Tea (Niaowang) in Southwest China and Their Relationship with Trace Elements

**DOI:** 10.3390/toxics11060502

**Published:** 2023-06-02

**Authors:** Yongcheng Jiang, Zhenming Zhang, Jiachun Zhang

**Affiliations:** 1College of Resources and Environment Engineering, Guizhou University, Guiyang 550025, China; gs.ycjiang22@gzu.edu.cn; 2Guizhou Institute of Biology, Guizhou Academic of Science, Guiyang 550009, China; 3Guizhou Karst Environmental Ecosystems Observation and Research Station, Ministry of Education, Guiyang 550025, China; 4Key Laboratory of Karst Georesources and Environment, Ministry of Education, Guizhou University, Guiyang 550025, China; 5Guizhou Botanical Garden, Guizhou Academy of Sciences, Guiyang 550004, China; gs.jczhang@gzu.edu.cn

**Keywords:** catechin, trace element, correlation, karst, Niaowang

## Abstract

This study investigated the relationship between the characteristics of quality components and trace elements of Niaowang tea from Guizhou Province in mountainous plateau areas. The contents of catechin monomers and eight other trace elements were measured using high-performance liquid chromatography (HPLC) and inductively coupled plasma mass spectrometry (ICP-MS), respectively. The results showed that the tender summer leaves of Niaowang tea in Guizhou Province had the highest content of catechins at 3558.15~2226.52 μg·g^−1^. The content of ester catechins was the highest in summer, amounting to 69.75~72.42% of the total catechins. The content of non-ester catechins was the highest in autumn, reaching 52.54~62.28% of the total catechins; among ester catechins, the mass fraction of epigallocatechin gallate (EGCG) showed a pattern of mature summer leaves > tender summer leaves > mature autumn leaves > tender autumn leaves, and the mass fractions of gallocatechin gallate (GCG) and epicatechin gallate (ECG) were larger in autumn than in summer; gallocatechin (GC) had no significant correlation with different trace elements, and Mn had no significant correlations with different catechin monomers. EGCG was significantly negatively correlated with As, Se, Hg, Pb, Ni, and Zn. Additionally, gallic acid (GA) was significantly negatively correlated with As, Hg and Ni. Other catechin monomers were largely significantly positively correlated with trace elements. The biochemical indicators of the phenotype of Niaowang tea show that the summer and autumn buds are suited for making high-quality green tea.

## 1. Introduction

Tea is widely consumed and recognized as one of the leading non-alcoholic beverages worldwide [1]. Guizhou Province in China plays a crucial role in the tea industry, significantly contributing to the region’s economic and social development [2]. Guizhou Niaowang tea, originating from Southwest China, is particularly remarkable and highly valued as a local tea variety. This tea displays exceptional characteristics, with abundant foliage, green buds, and velvety trichomes [3]. The polyphenols in tea, particularly catechin-like substances, are of utmost importance as they constitute a class of essential bioactive compounds [4]. Catechins make up 60–80% of the total polyphenols in tea and significantly impact its color, aroma, and flavor [5]. Recent research has clearly revealed multiple benefits of catechins for the human body, especially the capacity of EGCG, a prominent catechin, to counteract free radicals through reactions that produce stable compounds [6]. Additionally, GCG has shown potential intervention capabilities in cardiovascular diseases [7,8]. The quality of tea leaves is intricately linked to their chemical composition, revealing significant variations across seasons, tea varieties, and cultivation techniques [9,10].

Guizhou Province, located in China’s karst region, occupies a prominent position due to its high background levels of soil trace elements [11,12,13]. The distinctive geological composition of the karst region results in the release of trace elements from rocks into the soil through processes of weathering and leaching [14,15]. Additionally, industrial and agricultural activities in the area contribute to varying degrees of excess trace elements in the soil that can be transferred to crops such as tea and subsequently absorbed by humans [16,17]. Extensive studies have revealed the adverse health effects associated with excessive levels of specific trace elements. For instance, elevated concentrations of Cd have been linked to impaired pancreatic secretion [18] and Pb has been associated with reproductive harm [19]. Similarly, As has the potential to cause damage to internal organs [20] and increased Zn levels have been implicated in raising the risk of Alzheimer’s disease in adults [21]. Furthermore, the production process of tea typically involves the direct processing of fresh tea leaves, potentially resulting in higher levels of contaminant residues compared with other crops. Previous research has shown that excessive levels of trace elements in soil not only adversely affect tea growth but also accumulate in tea through root systems [22]. Tea plants, in particular, have a remarkable capacity for accumulating trace elements [23]. In Guizhou Province’s Guiding, Duyun, and Meitan counties, the enrichment coefficients of Cd, Hg, Cu, and Pb in tea have been observed to range from 5.57% to 27.7%. Considering these circumstances, conducting research on soil trace elements and tea catechins in Guizhou Province is of utmost importance [24]. Such investigations can offer vital insights into the potential risks associated with trace element contamination in tea production and consumption, thus ensuring the safety and quality of tea products in the region.

Currently, the academic community has extensively focused on trace elements in tea, with a particular emphasis on their characterization [25], health risk assessment [26], enrichment patterns [27], and leaching patterns [28]. However, studies exploring the interrelationships between biochemical components and trace elements in tea are relatively limited. This study aimed to analyze eight catechin monomers in tea, focusing on identifying content differences and examining their correlations with trace elements, especially during the summer and autumn seasons. By doing so, the study aimed to generate fundamental data that will contribute to the effective cultivation and management of tea plantations and facilitate the control of tea quality in the karst mountains of Southwest China.

## 2. Materials and Methods

### 2.1. Overview of Researched Region

The research area is located in Guiding County, Guizhou Province. The county, situated in the center of the mountainous area of Central Guizhou, covers an area of 107°08′–107°15′ E and 26°40′–26°47′ N, with an average elevation of 1000–1300 m. The region exhibits a humid central subtropical monsoon climate, with an annual average temperature, precipitation, and frost-free period of 15 °C, 1143 mm, and 289 days, respectively. In essence, the region is characterized by well-defined seasons, suitable temperatures, abundant precipitation, and an extended frost-free period. The geomorphological types are mainly mountainous areas, followed by hills and a few dam fields. The soil types are mainly yellow soil, followed by limestone and paddy soil, as well as purple and flu-aquic soil in some local areas.

### 2.2. Sample Collection and Preparation

A total of 56 samples of Niaowang tea were collected in Guiding County, Guizhou Province, as shown in Table 1. The collection of samples was carried out considering the different seasons when the tea was picked. For instance, summer tea corresponds to tea harvested from early June to early July, while autumn tea represents tea harvested between mid-autumn and late September. Samples of summer and autumn tea in this study were collected on 15 June and 15 September, respectively. During sampling periods, the number of sprouting twigs, the length of new twigs, and budding density were observed and measured at the sampling sites. The collected tea was dried to constant weight in a constant-temperature drying oven at 120 °C. The dried samples were ground for 1 min with a high-speed grinder and sieved to pass a 40–60 mesh.

### 2.3. Measurement of Tea Quality and Trace Elements

High-performance liquid chromatography (HPLC) was adopted to measure the 8 components of the catechin compounds, catechin (C), epicatechin (EC), epigallocatechin (EGC), gallocatechin (GC), gallocatechin gallate (GCG), epigallocatechin gallic acid (EGCC), epicatechin gallate (ECG) and gallic acid (GA). A 0.2 g tea powder sample was accurately weighed (accurate to 0.0001 g) and put into a 10 mL centrifuge tube. Then, 5 mL of a 70% methanol solution prewarmed at 70 °C was added to the tube. The mixture underwent water bath extraction at 70 °C for 10 min and then let cool to room temperature, followed by centrifugation for 10 min. The resulting supernatant was removed and placed into a 10 mL volumetric flask, and the residues were extracted again using a 5 mL 70% methanol solution. The resulting solutions were merged to a fixed volume of 10 mL, which was filtered with a 0.45 um microporous membrane. A 2 mL extraction solution was mixed with an 8 mL stabilizing solution, which was then filtered with a 0.45 um membrane and subjected to an HPLC test (HPLC-20AT, Shimadzu Corp, Shanghai, China). The measurement parameters were as follows: C18 columns, flow rate of 1 mL/min, column temperature of 35 °C, detection wavelength of 278 nm, and gradient elution. The results were directly quantified with the external standard method of catechins [29].

The contents of trace elements in tea were measured with inductively coupled plasma mass spectrometry (ICP-MS, Agilent 7500 a). A 1 g sample was weighed and put into a high-pressure sealed vessel, to which 5 mL of a nitric acid solution and 2 mL of hydrogen peroxide were added. Digestion was performed at 250 °C, 400 °C, and 450 °C for 6 min, 5 min, and 5 min, respectively. After the digestion tank cooled down, the solution was removed, placed into a 50 mL volumetric flask, and rinsed 3 times with doubly deionized water before the obtainment of a constant volume of sample solution to be measured. The sample was measured with an inductively coupled plasma source mass spectrometer [30].

### 2.4. Data Processing

SPSS 2022 (IBM, Armonk, NY, USA) was used to analyze the significance of differences. Multiple comparisons for difference analysis were carried out with the LSD method. *p* < 0.05 was considered statistically significant. Origin 2021 (Origin Lab, Northampton, MA, USA) was applied to conduct principal component analysis and develop correlation heat maps.

## 3. Results

### 3.1. Analysis of Differences in Total Catechins between Summer and Autumn Niaowang Tea

The measured results of the total catechins (the combined content of the measured catechin components), ester-type catechins, and non-ester-type catechins in Guizhou Niaowang tea, categorized by season and maturity level, are presented in Table 2. The total catechin contents in the tender and mature summer leaves were significantly higher than their autumn counterparts, with the former reaching 3558.15 μg·g^−1^ and 2226.52 μg·g^−1^, respectively, amounting to 0.35% and 0.22% of the total tea mass and 1.45 and 3.6 times of those of their autumn counterparts. Catechins in tea can be divided into ester catechins and non-ester catechins. The content of ester catechins was the highest in tender summer leaves, reaching 2469.43 μg·g^−1^ and representing 69.74% of the total catechins; the same content was the lowest in mature autumn leaves, reaching 423.40 μg·g^−1^ and representing 47.45% of the total catechins. The content of non-ester catechins was the highest in tender autumn leaves and the lowest in mature autumn leaves. Generally, the ester catechins in Niaowang tea were found to significantly vary across seasons, with their contents higher in summer and lower in autumn. Comparatively, the contents of non-ester catechins were found to be irrelevant to seasons but relevant to maturity level. In the same seasonal condition, the contents of non-ester catechins in tender leaves were larger than those in in mature leaves.

### 3.2. Analysis of Differences in Contents of Catechin Monomers in Summer and Autumn 

The ester catechins in the tea mainly consisted of EGCG, GCG and ECG. As can be seen in Figure 1, the mass fraction of EGCG was the largest in the studied tea, accounting for 60.42–67.87% in summer and 27.37–37.57% in autumn. The magnitude of the mass fraction exhibited a gradient change in the form of mature summer leaves > tender summer leaves > mature autumn leaves > tender autumn leaves. The distribution of the mass fraction of GCG was exactly the opposite of that of EGCG, with the former being tender autumn leaves > mature autumn leaves > tender summer leaves > mature summer leaves. The mass fraction of ECG was the lowest in mature summer leaves, with no significant differences under other seasonal or maturity conditions. The non-ester catechins mainly consisted of C, EC, EGC, GC and GA. The mass fractions of C and EC in autumn were significantly higher than those in summer. The mass fraction of EGC was the highest in mature autumn leaves, reaching 19.60%, and lowest in tender summer leaves, reaching 5.36%. The mass fraction of GC was the highest in tender autumn leaves and the lowest in mature summer leaves, but it showed no significant difference in tender summer leaves and mature autumn leaves. The mass fraction of GA was 1.14% in tender summer leaves and 0.16% in tender autumn leaves, representing an 8.14-fold difference.

### 3.3. Principal Component Analysis of Catechins in Niaowang Tea

A principal component analysis of the catechin contents in 56 Niaowang tea resources was conducted using the Origin 2021 software, as shown in Table 3. The cumulative contribution rate of the first three principal components reached 81.73%, indicating that these components included the majority of information on all traits and could be used to comprehensively assess the materials. The contribution rate of the first principal component that reflects information on ester catechins was 47.50%, with EGCG making the largest contribution. The contribution rate of the second principal component was 21.42%, with EGC making the largest contribution. The contribution rate of the third principal component was 12.80%, with GC making the largest contribution. Both the second and third principal components reflect the information on non-ester catechins.

As shown in Figure 2, the 56 Niaowang tea resources were distributed in the I–IV quadrants of the principal component space. Specifically, tender summer leaves were all distributed in quadrant II. Mature summer leaves were all distributed in quadrant III. Tender autumn leaves were mainly distributed in quadrants I and IV, with the majority found in quadrant I. Mature autumn leaves were distributed in quadrants I, III and IV, with the majority found in quadrant IV. The distributions of the 56 tree plant resources in the principal component space indicate substantial differences in terms of the contents of eight catechins in Niaowang tea across seasons and maturity levels. The loading diagram shows the key indicators causing the differences in the 56 Niaowang tea samples across seasons and maturity levels. The positions of GA and EGCG in the loading plot are similar to those of tender summer leaves in the score chart, indicating that compared with other seasons and maturity levels, these two catechin monomers were highly concentrated in the tender summer leaves. The positions of ECG, EC and GC in the loading plot present opposite patterns to those in the mature summer leaves but similar to those of most tender autumn leaves in the score chart. These results indicate that the contents of these three catechins were relatively lower in the mature summer leaves but higher in the tender autumn leaves. The positions of GCG, C and EGC in the loading plot are similar to those of the mature autumn leaves in the score chart, indicating relatively higher contents of these three catechin monomers in the mature autumn leaves. These results were consistent with the data of the component matrix.

### 3.4. Trace Element Analysis in Niaowang Tea Leaves

The testing and analysis results of the trace elements in the 56 samples of Niaowang tea in Guizhou Province are shown in Table 4. The trace element contents in the tea significantly varied with seasonality, with higher contents in autumn than in summer. Moreover, the trace element variations in the eight tea leaves were also large, with the contents ranging from As < Hg < Se < Pb < Ni < Zn < Mn in ascending order. The lowest was As, with mean values of 0.02 mg·kg^−1^ in summer and 0.28 mg·kg^−1^ in autumn, and the highest was Mn, with mean values of 250 mg·kg^−1^ in summer and 423 mg·kg^−1^ in autumn. The difference between them could be up to four orders of magnitude.

### 3.5. Analysis of Correlations between Catechin Monomers and Trace Elements 

The correlations between catechin monomers and trace elements in Niaowang tea are shown in Figure 3, where positive and negative correlations are denoted by red and blue, respectively, and deeper colors represent higher correlation levels. As can be seen in the figure, Mn had no significant correlations with different catechin monomers, GC presented no significant correlations with trace elements, and GA was negatively correlated with As, Hg and Pb but had no significant correlations with other trace elements. EGC had extreme significantly positive correlations with As, Se and Pb, and C also had an extreme significantly positive correlation with Ni, but neither of them showed significant correlations with other trace elements. EC and GCG both had significantly positive correlations with trace elements and extreme significantly negative correlations with EGCG. EGCG showed significantly negative correlations with the measured trace elements. These results indicate that the contents of trace elements may be relatively lower in EGCG-rich tea leaves, offering a potential clue for finding EGCG-rich tea plant resources. ECG showed extreme significantly positive correlations with Ni, Pb, Se and Zn but no significant correlations with Mn, As and Hg. The contents of As and Hg in the tea did not exceed the national limit value, but the average value of Pb exceeded the national limit value, indicating that the contents of As and Hg in the soil in the study area may be relatively low but the content of Pb is relatively high.

## 4. Discussion

This study investigated the total catechin contents of Niaowang tea, revealing distinct seasonal variations. Higher levels were observed during summer (range: 2226.52 to 3558.15 μg·g^−1^), while lower levels were found during autumn (range: 912.07 to 2454.02 μg·g^−1^). These findings were supported by Tang’s [31] research on loquat tea in Xing Ying, Sichuan, which also showed higher catechin contents during summer (range: 131.64 to 158.28 mg·g^−1^) and lower contents during autumn (range: 122.11 to 144.48 mg·g^−1^). While Niaowang tea exhibited lower catechin contents compared with other tea varieties in this study, the overall trend remained consistent. Jin’s [32] trials in Hubei further confirmed these results. These findings imply regional variations in catechin contents, indicating the impact of environmental factors on tea cultivation. Specifically, catechin contents exhibit positive correlations with temperature, light conditions, and average annual evapotranspiration [33,34] but show negative correlations with soil pH in tea plantations [35]. Compared with Sichuan and Hubei, Guizhou Province has a lower average temperature, fewer light hours, and reduced evaporation. Preliminary investigations indicated that despite the substantial use of chemical fertilizers during tea cultivation in Guizhou, inadequate absorption by tea trees and excessive application has led to reduced soil pH in tea plantations. This observation may provide an explanation for the reduced catechin contents in Guizhou Niaowang tea. These findings highlight the importance of considering environmental factors, such as temperature, light conditions, evaporation, and soil pH, in tea cultivation to optimize catechin contents. Striking a balance between fertilizer usage to ensure adequate nutrient availability and maintaining an appropriate soil pH is crucial for achieving optimal levels of tea catechins (Appendix A).

In recent years, there has been growing scholarly interest in trace element contamination and its effects, particularly the accumulation of trace elements in crops and the associated health risks. Studies conducted by Wen [36] and Han [37] investigated the characteristics and origins of trace elements in soils and crops in Guizhou and Southern Sichuan. Their findings revealed elevated levels of soil trace elements in both regions compared with most provinces in China. Additionally, the trace element levels in tea plantations in Guizhou were found to be higher than those in Southern Sichuan Province. Moreover, He [38] identified the most prominent enrichment effect of Zn in crops within the agricultural region of Guizhou Province. This effect was primarily attributed to the consistently high background levels of soil trace elements in Guizhou Province. The enrichment of trace elements was found to influence the levels and activity of tea catechins, particularly EGCG. Studies have demonstrated that elevated levels of Hg and Pb exhibit a greater binding capacity to catechins [39] and increased concentrations of Mn decrease the antioxidant activity of EGCG [40]. Moreover, a lower pH and the presence of trace elements facilitates the acidification of EGCG and diminishes the catechin content in tea [41]. Consequently, elevated levels of trace elements have an adverse impact on the accumulation of catechins in tea. This aligns with our findings indicating a negative correlation between EGCG and most catechins in tea. Our results also showed that trace elements were lower in summer when catechin levels were higher and higher in autumn when catechin levels were lower, while the findings of a negative correlation between EGCG and trace elements in tea were consistent.

Catechins serve as the primary astringent compounds in tea and offer various health benefits including antioxidant, anticancer, cardioprotective, and lipid metabolism modulation properties. The correlation between catechins and trace elements in tea plants during different seasons suggests that trace elements influence the accumulation of catechins; however, the mechanistic understanding of this influence is not comprehensive and warrants further exploration.

## 5. Conclusions

The contents of catechins in Guizhou Niaowang tea were found to be 3558.15~2226.52 μg·g^−1^ and 2454.02~912.07 μg·g^−1^, respectively, in summer and autumn. The content of ester catechins was higher and that of non-ester catechins was lower in summer. The content of non-ester catechins was higher in tender tea leaves and lower in mature leaves. Ester catechins in Niaowang tea were found to significantly vary across seasons, with their contents being higher in summer and lower in autumn. Comparatively, the contents of non-ester catechins were found to be irrelevant to season but relevant to maturity level. In the same seasonal condition, the contents of non-ester catechins in tender leaves were larger than in mature leaves. Mn did not have a significant correlation with any of the catechin monomers; EGCG was significantly negatively correlated with As, Hg, Se, Pb, Ni, and Zn; and GA was significantly negatively correlated with As, Hg and Ni. Other catechin monomers were largely significantly positively correlated with trace elements.

## Figures and Tables

**Figure 1 toxics-11-00502-f001:**
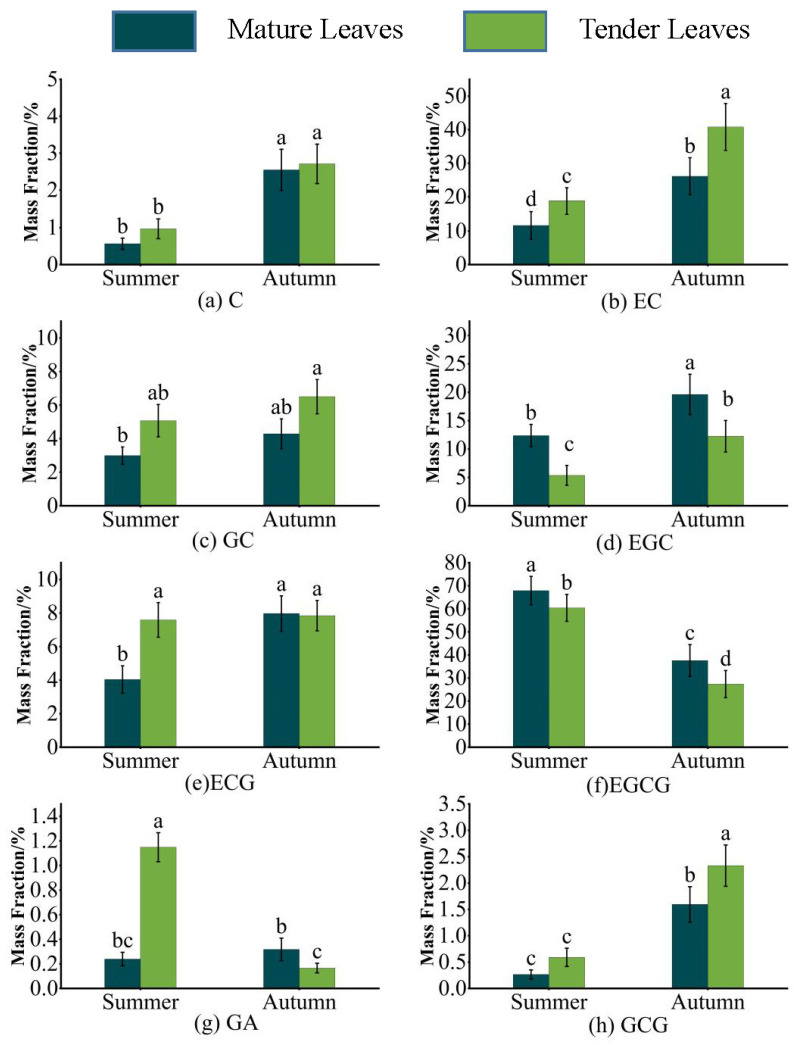
Comparison of 8 catechin monomers in Niaowang tea. Different letters in the same graph represent significant differences (*p* < 0.05).

**Figure 2 toxics-11-00502-f002:**
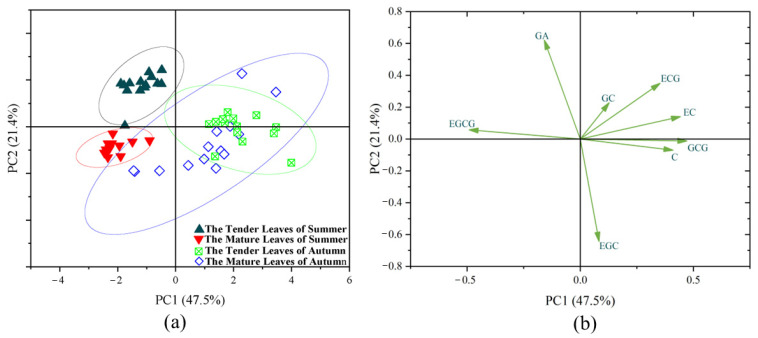
Principal component analysis of 56 Niaowang tea resources: (**a**) principal component score diagram; (**b**) load diagram.

**Figure 3 toxics-11-00502-f003:**
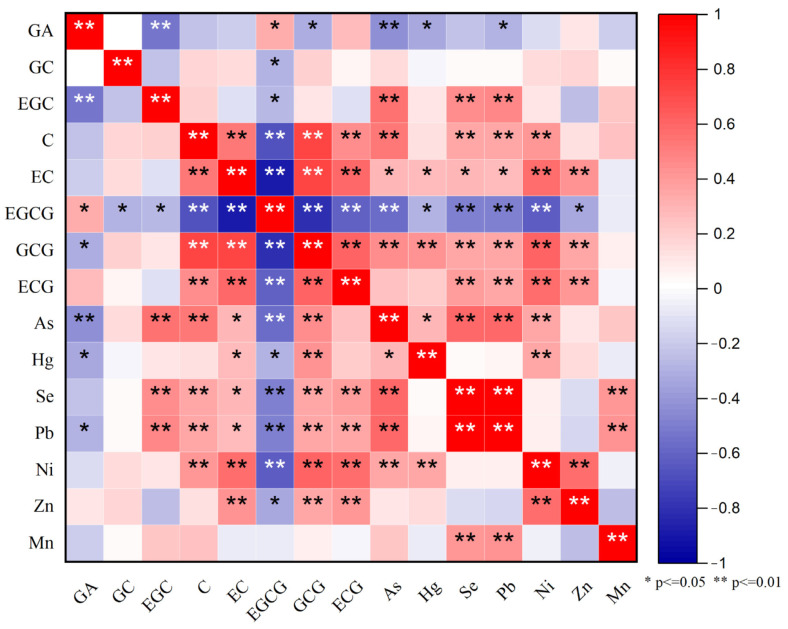
Heat map of catechin monomers and trace element correlations. * Represents correlation at the 0.05 level; ** represents correlation at the 0.01 level.

**Table 1 toxics-11-00502-t001:** Information on sampling points in the study area.

Sample Points	Mature Leaves Number	Tender Leaves Number	Latitude and Longitude	Slope	Parent Material
Shangba town	8	8	107°01.811 E, 26°12.564 N	25–40	Sticky yellow soil
Gongchabei	2	2	107°01.584 E, 26°11.916 N	15–30	Sand shale
Gaozhai town	2	2	107°01.603 E, 26°12.042 N	10–40	Sandstone
Nongye station	2	2	107°03.060 E, 26°13.518 N	15–30	Sticky yellow soil
Shuili station	6	6	107°03.437 E, 26°13.656 N	15–40	Sticky yellow soil
Fangjia	6	6	107°03.609 E, 26°13.732 N	10–30	Sandstone
Tiechang	2	2	107°06.433 E, 26°13.217 N	15–25	Sticky yellow soil

**Table 2 toxics-11-00502-t002:** Total catechin, ester-type catechin, and non-ester-type catechin contents in Niaowang tea leaves in summer and autumn (μg/g).

	Summer		Autumn	
	Tender Leaves	Mature Leaves	Tender Leaves	Mature Leaves
Total catechins	3558.15 ± 841.81 a	2226.52 ± 291.50 b	2454.02 ± 845.40 b	912.07 ± 623.89 c
Ester-type catechins	2469.43 ± 721.82 a	1609.30 ± 260.60 b	905.61 ± 315.82 b	423.40 ± 278.78 c
Non-ester-type catechins	1088.71 ± 210.19 a	617.16 ± 139.51 b	1548.41 ± 602.06 a	488.67 ± 356.63 b

Different letters in the same row represent significant differences (*p* < 0.05).

**Table 3 toxics-11-00502-t003:** Principal component analysis of biochemical tea component indexes.

Variable	Principal Component		
	PC1	PC2	PC3
EGCG	−0.96	0.08	0.05
GCG	0.92	−0.02	0.01
EC	0.86	0.18	0.05
C	0.80	−0.09	0.01
ECG	0.69	0.46	0.39
EGC	0.16	−0.84	0.17
GA	−0.31	0.81	0.27
GC	0.25	0.30	−0.87
Eigenvalue	3.80	1.71	1.02
Percentage of Variance (%)	47.51	21.42	12.80
Cumulative (%)	47.51	68.93	81.73

**Table 4 toxics-11-00502-t004:** Trace element contents in Niaowang tea leaves (mg·kg^−1^).

	Summer	Autumn
As	0.02 ± 0.01	0.28 ± 0.13
Hg	0.12 ± 0.04	0.25 ± 0.15
Se	0.14 ± 0.06	0.97 ± 0.23
Pb	1.25 ± 0.34	13.43 ± 3.25
Ni	3.55 ± 1.23	8.86 ± 3.22
Zn	23.66 ± 6.73	30.07 ± 8.36
Mn	250.3 ± 55.47	423.43 ± 120.35

## Data Availability

The datasets generated and analyzed during the current study are available from the corresponding author on reasonable request.

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
