# Peer review of "Quality Characteristics of Karst Plateau Tea (Niaowang) in Southwest China and Their Relationship with Trace Elements"

_toxics, 2023, doi:10.3390/toxics11060502_

Round 1

Reviewer 1 Report

The study on the topic "Quality characteristics of karst plateau tea in Southwest China 2 and their relationship with trace elements" is of great interest, revealing the main patterns of the formation of tea as a cultivated plant, as well as one of the most popular product in the food industry market. The study is aimed at studying the relationship between the quality characteristics of tea and the content of trace elements. The work can be accepted for printing after making some changes.

The text lacks references to tables and figures, which significantly complicates the perception of information.

It is recommended to rename table #2. The current name "Proportion of ester-type catechins and non-ester-type catechins" does not reflect all the data given in it, since the table contains not only relative, but also absolute values indicating the maturity of the leaves. All this should be indicated in the title. The same table shows the letters (a, b). You should provide the decoding of the letters in the footnote to the table.

Page 145. This sentence does not require an explanation in brackets "(as a percentage)"

The designations in Figure 2 are not readable; the type is too small. Please increase.

Page 204-206. This explanation of the figure is optional and can be removed.

The paragraph discussing the results should be substantially improved. The first two sentences (pp. 226-230) do not contain general information on the research topic and are more related to the introduction. The text of the discussion should not only provide information about the results of similar studies. This section can be improved by comparing the results of the current study with previous work, as well as by explaining the results obtained using previous research results on related topics.

Why is the table with the content of trace elements in tea leaves placed in a separate application?

The English needs a minor improving

Author Response

Firstly, thanks a lot to the contribution by the editors and reviewers. All the comments are precious and indispensable for the perfect of this manuscript and will help to improve the importance of our research.

Based on the comments of reviewers and your suggestions, I have carried out the corresponding revision for my paper. The replies to referees are as follow one by one. In addition, all the amendments in this revised paper are provided with a red font so that you could check clearly.

We have revised the manuscript based on the comments from editors and reviewers as followed:

1The text lacks references to tables and figures, which significantly complicates the perception of information.

Reply: Thanks a lot to the contribution by the reviewer. We have carefully checked all the figures and tables and made references to all the tables and figures in the text.

2It is recommended to rename table #2. The current name "Proportion of ester-type catechins and non-ester-type catechins" does not reflect all the data given in it, since the table contains not only relative, but also absolute values indicating the maturity of the leaves. All this should be indicated in the title. The same table shows the letters (a, b). You should provide the decoding of the letters in the footnote to the table.

Reply: Thank you very much for your suggestions. We have modified the table to remove the percentages (see table2), and added the decoding of the letters in all tables and figures.

3Page 145. This sentence does not require an explanation in brackets "(as a percentage)"

Reply: Thanks a lot to the contribution by the reviewer. We have removed this sentence from the text.

4The designations in Figure 2 are not readable; the type is too small. Please increase.

ReplyThank you for drawing attention to the section. After careful and prudent consideration, we believe that your proposal is very correct, so we have made the graphic description fonts larger.

5Page 204-206. This explanation of the figure is optional and can be removed.

Reply: Thanks a lot to the contribution by the reviewer. we have deleted the figure and the explanation of the figure has been removed.

6The paragraph discussing the results should be substantially improved. The first two sentences (pp. 226-230) do not contain general information on the research topic and are more related to the introduction. The text of the discussion should not only provide information about the results of similar studies. This section can be improved by comparing the results of the current study with previous work, as well as by explaining the results obtained using previous research results on related topics.

Reply: We would like to express our appreciation for your comment. After deliberating on your comments. We have extensively revised the article by adding a comparison of experimental data for catechins in the first paragraph (lines 244-252) and also explaining the possible reasons for the discrepancies in the data (lines 252-262). The correlation between tea catechins and trace elements was explained in the second paragraph (lines 263-283), and the emphasis of the next study was discussed in the third paragraph (lines 284-289).

7Why is the table with the content of trace elements in tea leaves placed in a separate application?

ReplyThanks a lot to the contribution by the reviewer. We initially thought of this table as a source of data for the figure3, but after careful consideration of your suggestions, we thought it would be better to present this table directly in the text to see the results more directly.

If you have any question about this paper, please don’t hesitate to let me know.

Thank you and all the referees very much for the kind advice.

Sincerely yours,

Zhenming Zhang

Reviewer 2 Report

Quality characteristics of karst plateau tea in Southwest China and their relationship with trace elements

Yongcheng Jiang, Zhenming Zhang, Jiachun Zhang

The article discusses the link between quality components and trace elements of Niaowang Tea estimated by HPLC and the variation of the components with changing climatic conditions. The presence of trace elements with tea constituents poses health hazards for humans. Although the authors discuss the benefits of tea constituents and their link with trace elements, they did not discuss the adverse health implications of trace elements.

What research efforts have been done to solve this problem? Discuss with examples.

 Tea polyphenols mainly include 4 types of substances, namely, catechins, flavonoids, anthocyanins, and phenolic acids, why other components were not measured?

What are the health-promoting effects of different tea constituents? Discuss briefly.

How does the present research aim to solve the problem associated with trace elements with tea components? What is the advantage of the present research in Guizhou Province?

The manuscript needs to be extensively revised for English language and typological errors.

Line 2: karst plateau tea is mentioned while in line 11: Niaowang Tea, is it the same tea? Be consistent.

Abbreviations: ICP-MS, ECCG, GCG and ECG, GC, GA use full forms at the first mention, then abbreviation subsequently in the text. Please revise.

Discussion, Line 226: Authors Catechin components are important secondary metabolites, please revise this sentence.

References can be improved.

The paper has to be thoroughly revised for the English language, and typographical errors. 

Author Response

Firstly, thanks a lot to the contribution by the editors and reviewers. All the comments are precious and indispensable for the perfect of this manuscript and will help to improve the importance of our research.

Based on the comments of reviewers and your suggestions, I have carried out the corresponding revision for my paper. The replies to referees are as follow one by one. In addition, all the amendments in this revised paper are provided with a red font so that you could check clearly.

We have revised the manuscript based on the comments from editors and reviewers as followed:

1、The article discusses the link between quality components and trace elements of Niaowang Tea estimated by HPLC and the variation of the components with changing climatic conditions. The presence of trace elements with tea constituents poses health hazards for humans. Although the authors discuss the benefits of tea constituents and their link with trace elements, they did not discuss the adverse health implications of trace elements.

Reply: Thanks a lot to the contribution by the reviewer. We have added the adverse effects of trace elements on human health in lines 50-56. For instance, elevated concentrations of Cd have been linked to impaired pancreatic secretion, and Pb has been associated with reproductive harm. Similarly, As has the potential to cause damage to internal organs, while increased Zn levels have been implicated in raising the risk of Alzheimer's disease in adults.

2What research efforts have been done to solve this problem? Discuss with examples.

ReplyWe would like to express our appreciation for your comment. After careful consideration of your comments, we have added the current research on trace elements and tea quality to the text(lines 67-71).

3Tea polyphenols mainly include 4 types of substances, namely, catechins, flavonoids, anthocyanins, and phenolic acids, why other components were not measured?

Reply: Thank you for drawing attention to this section. The main antioxidant and active ingredient in tea is catechins, which accounts for 60-80% of tea polyphenols. Initially, we mentioned these four ingredients to make the narrative more complete and to avoid giving the impression that tea polyphenols are catechins. However, after careful consideration of your suggestion, we felt that we should not mention the other ingredients in the text to avoid controversy, so we have removed them from the text. Of course, considering the effects of other ingredients is also the emphasis of our next research. (Lines 34-36).

4What are the health-promoting effects of different tea constituents? Discuss briefly.

ReplyThank you for drawing attention to the section. After careful consideration of your comments, we think it is necessary to add the benefits of catechins in tea to the body in the text. Therefore, we added in the text the important role of catechin components in promoting health in human body. For instance, EGCG can clear free radicals in human body and has a great impact on the antioxidant capacity of human body, and also GCG has some protective effect on human diseases (Lines 36-41).

5How does the present research aim to solve the problem associated with trace elements with tea components? What is the advantage of the present research in Guizhou Province?

ReplyWe would like to express our appreciation for your comment. We provide a detailed description of the research implications of our study in lines 64-66. In the introduction of our paper, we have emphasized that Guizhou province's karstic geological background results in the introduction of trace elements into the soil through rock weathering. Additionally, the emissions from industrial and agricultural activities have led to higher background values of trace elements in Guizhou compared to other provinces in China (Lines 44-50). Considering the significance of tea as an economic industry in Guizhou, particularly the importance of catechin as a key ingredient in tea, it is crucial to investigate the relationship between trace elements and catechin in this high-background trace element region. Such research can greatly contribute to the effective cultivation and management of tea plantations in Guizhou province. Furthermore, the findings can serve as a valuable reference for tea cultivation and management in other areas with similar high background levels of trace elements (Lines 73-76).

6The manuscript needs to be extensively revised for English language and typological errors.

ReplyThank you for drawing attention to the section. We have revised the grammar and punctuation of the full text, and have asked senior experts of our profession to help check it. Thanks!

7Line 2: is mentioned while in line 11: Niaowang Tea, is it the same tea? Be consistent.

ReplyThanks a lot to the contribution by the reviewer. We have indicated Niaowang in parentheses after karst plateau tea in the title to avoid readers thinking that they are two different kinds of tea. (Line 2)

8Abbreviations: ICP-MS, ECCG, GCG and ECG, GC, GA use full forms at the first mention, then abbreviation subsequently in the text. Please revise.

ReplyThanks a lot to the contribution by the reviewer. We have used the full form of the abbreviation when it is first mentioned in the text and have indicated the abbreviation in parentheses (Lines 12-13 and 18-24).

9Discussion, Line 226: Authors Catechin components are important secondary metabolites, please revise this sentence.

Reply: We would like to express our appreciation for your comment. Due to our mistake, this error was caused, and we have performed a deletion on this sentence.

10References can be improved.

Reply: Thanks a lot to the contribution by the reviewer. We have carefully reviewed the references and then modified the reference format according to the journal requirements.

If you have any question about this paper, please don’t hesitate to let me know.

Thank you and all the referees very much for the kind advice.

Sincerely yours,

Zhenming Zhang
